# The Psychological Impact of COVID-19 among Women Accessing Family Care Centers during Pregnancy and the Postnatal Period in Italy

**DOI:** 10.3390/ijerph19041983

**Published:** 2022-02-10

**Authors:** Ilaria Lega, Alessandra Bramante, Laura Lauria, Pietro Grussu, Valeria Dubini, Marcella Falcieri, Maria Carmen Ghiani, Antonia Giordano, Stefania Guidomei, Anna Domenica Mignuoli, Serena Paris, Maria Enrica Bettinelli, Patrizia Proietti, Silvia Andreozzi, Valeria Brenna, Mauro Bucciarelli, Gabriella Martelli, Claudia Ferraro, Melissa Torrisi, Danilo Carrozzino, Serena Donati

**Affiliations:** 1National Centre for Disease Prevention and Health Promotion, Istituto Superiore di Sanità—Italian National Institute of Health, Viale Regina Elena 299, 00161 Rome, Italy; laura.lauria@iss.it (L.L.); silvia.andreozzi@iss.it (S.A.); mauro.bucciarelli@iss.it (M.B.); gabriella.martelli@iss.it (G.M.); claudia.ferraro@iss.it (C.F.); serena.donati@iss.it (S.D.); 2Società Marcé Italiana per la Salute Mentale Perinatal, Italian Marcé Society, Via Donatello 26A, 20131 Milan, Italy; alessandra.bramante@gmail.com (A.B.); valeria.brenna@hotmail.it (V.B.); 3Consultorio Familiare South Padua District, Azienda ULSS 6 Euganea, Via Enrico degli Scrovegni 14, 35131 Padua, Italy; pietro.grussu@aulss6.veneto.it; 4Area Assistenza Sanitaria Territoriale e Continuità, Azienda USL Toscana Centro, Piazza Santa Maria Nuova 1, 50122 Firenze, Italy; valeria.dubini@uslcentro.toscana.it; 5AUSL di Bologna, Via Castiglione, 29, 40124 Bologna, Italy; marcella.falcieri@ausl.bologna.it (M.F.); stefania.guidomei@ausl.bologna.it (S.G.); 6ASSL di Olbia, ATS Sardegna, Via Bazzoni-Sircana 2/2°, 07026 Olbia, Italy; mariacarmen.ghiani@atssardegna.it; 7Attività Consultoriali ASL TO3, Via Martiri XXX Aprile 30, 10093 Collegno, TO, Italy; antogiordano@aslto3.piemonte.it; 8ASP Cosenza, Viale Degli Alimena 8, 87100 Cosenza, Italy; anna.mignuoli@regione.calabria.it; 9Direzione Professioni Sanitarie e Sociali, ASST Bergamo Est, Via Paderno 21, 24068 Seriate, BG, Italy; serena.paris@asst-bergamoest.it; 10Coordinamento Attività Consultoriali, ASST Fatebenefratelli Sacco, Via Sant’ Erlembardo 4, 20126 Milan, Italy; maria.bettinelli@unimi.it; 11UOC Assistenza Alla Persona, ASL Roma 2, Via Maria Brighenti 23, 00159 Rome, Italy; patrizia.proietti@aslroma2.it; 12Dipartimento di Medicina Clinica e Sperimentale, Università di Pisa, Via Savi 10, 56126 Pisa, Italy; melissatorrisi1@gmail.com; 13Department of Psychology “Renzo Canestrari”, University of Bologna, Viale Berti Pichat 5, 40127 Bologna, Italy; danilo.carrozzino@unibo.it

**Keywords:** maternity care, COVID-19, family care centres, mental health, psychological distress, Italy

## Abstract

There has been concern about the impact of the COVID-19 outbreak on women’s mental health during the perinatal period. We conducted a cross-sectional web-based study aimed at evaluating the psychological impact (BSI-18) of the COVID-19 pandemic on this population and collecting information on the perinatal experiences (COPE-IS) during the second Italian wave. Overall, 1168 pregnant women, and 940 within the first six months after childbirth, were recruited in selected Italian Family Care Centers from October 2020 to May 2021. The prevalence of psychological distress symptoms during pregnancy was 12.1% and 9.3% in the postnatal group. Financial difficulties, a previous mood or anxiety disorder and lack of perceived social support and of support provided by health professionals were associated to psychological distress symptoms in both groups. A third of the women felt unsupported by their social network; 61.7% of the pregnant women experienced changes in antenatal care; 21.2% of those in the postnatal period gave birth alone; more than 80% of the participants identified access to medical and mental health care and self-help as important resources in the present context. Health services should assure enhanced support to the most vulnerable women who face the perinatal period during the pandemic.

## 1. Introduction

The COVID-19 pandemic has had a considerable psychosocial impact, particularly among the most vulnerable groups of the population, including women in pregnancy and in the first year after childbirth [1,2]. Therefore, the need for evidence focusing on the indirect effects of the COVID-19 pandemic on the health of women in the perinatal period, including psychosocial aspects, and on measures to improve access and provision of care, including psychosocial assistance, has been authoritatively affirmed [3]. In a time of such significant biological and psychosocial changes, women in pregnancy and the postnatal period have had to face stressful factors, such as fear of contagion and possible risks for the fetus or the newborn, social isolation, the impossibility of receiving emotional and practical support from family and friends, changes in the relationship with their partner exacerbated by forced cohabitation, the limitation of face-to-face medical and obstetrical consultations, anxiety of giving birth without the support of the companion of choice and economic uncertainty due to job loss [2,4]. The well-established negative impact of maternal psychological distress on pregnancy and fetal and infant outcomes [5,6] call health services and professionals to action to minimize pandemic-related stress and promote maternal psychosocial well-being.

A systematic review of 81 studies that used validated measures to assess mental health outcomes in pregnant and postnatal women during the COVID-19 pandemic concluded that depressive and anxiety symptoms in perinatal women during the pandemic were higher compared to pre-pandemic values; that financial strain, decreased social and family support and low education were the key sociodemographic factors associated with increased depression and anxiety in perinatal women in the COVID outbreak and that adequate sleep, moderate physical activity and positive social support play a protective role, being negatively associated with perinatal symptoms of depression and anxiety in the pandemic context [7]. The main methodological limitations of the available evidence include the use of convenience sampling based on online recruitment of participants, which made it difficult to obtain truly representative samples, and the small sample sizes, due to the complexity and urgency of obtaining data during the pandemic [7].

Italy, the first European country in which cases of COVID-19 were registered, had the second largest number of COVID-19 infections after China and a high case-fatality rate in March 2020, while the Italian National Health Service (NHS) faced a massive burden [8]. The incidence of SARS-CoV-2 positive cases, the spread of the infection and the number of COVID-19 deaths did not have a homogeneous distribution throughout the national territory. In the first epidemic wave (end of February–May 2020, with a peak observed in March and April), the northern Regions were the most affected, while in the center and southern-insular area of the country, the epidemic had a lower impact [9]. On the contrary, during the second wave (October–December 2020) the virus spread was more homogeneous throughout the country [9]. A first phase of national lockdown (March–May 2020) was followed by the application of tiered measures involving limitations to retail and service activities, individual movement restrictions and reinforced distance learning in schools, which varied on a regional basis depending on incidence, time-varying reproduction number (Rt) and possible impact on the territory starting from November 2020 [10].

Perinatal care underwent major changes during the Italian pandemic emergency. Maternity services limited companion visits or access; women with confirmed SARS-CoV-2 infection, suspected cases and, sometimes, even women not affected by the virus gave birth alone during the pandemic’s first wave, and the practice of skin to skin contact after birth was often neglected [11]. Protecting childbirth physiology and preserving the mother–child bond turned into a national challenge [11].

The Family Care Centers (FCCs), the Italian NHS’s primary care services dedicated to family and women’s health, are about 1800 in number at the national level, with a ratio of 1:32.000 inhabitants, and they represent a unique reality at the international level [12]. Based on a multidisciplinary, proactive and holistic approach, the FCCs ensure pre- and postnatal care to mothers and their families, including maternal and fetal assessments, antenatal classes, breastfeeding promotion and postpartum support. During the pandemic, FCCs were forced to limit their activity to unpostponable services, while in-presence group perinatal activities were suspended all over the country [13]. Many FCCs have been exemplary in promptly reorganizing their care in the new context, enhancing phone counselling during pregnancy and breastfeeding already in March 2020 and providing online antenatal and postnatal classes from May 2020 [13].

The present multicenter study primarily aimed at evaluating the psychological impact of the COVID-19 pandemic on pregnant and postpartum women. We hypothesized that well-known socio-demographic and medical history risk factors for common mental disorders (i.e., low income, poor social support, previous history of mental disorders) and specific pandemic-related risk factors (i.e., previous SARS-CoV-2 infection, death of a loved one from COVID-19, residency in an area with a higher number of COVID-19 deaths) would be associated with higher psychological distress. A secondary aim was to collect information about the support that women received from FCCs during the pandemic and to analyze which further resources should be provided by these health centers, according to the users’ perspective.

## 2. Materials and Methods

### 2.1. Study Setting and Participants

This was a cross-sectional web-based study. The participants were recruited through the FCCs of nine Local Health Units located in eight Italian regions belonging to the northern (Piedmont, Lombardy, Veneto, Emilia-Romagna), central (Tuscany, Latium) and southern-insular (Calabria, Sardinia) Italian geographical macro-area from 1 October 2020 to 31 May 2021. The inclusion criteria were age of 18 years or above, being pregnant or in the first 6 months after childbirth and living in Italy. Eligible women were informed about the study by the FCCs’ health professionals during in-person or remote interactions and via health service social media and/or health service web sites; fliers describing the study were also available in the waiting rooms. Women willing to participate received the link to the web survey and, after providing informed consent to take part in the study, were enabled to complete the online questionnaire.

### 2.2. Instruments

The experiences of the participants during the COVID-19 pandemic were assessed through the Italian adaptation of the Coronavirus Perinatal Experiences Impact Survey (COPE-IS). The COPE-IS questionnaire, a new measure whose psychometric properties are yet to be established, was originally developed in English to assess several areas of impact of COVID-19 on women in pregnancy and the postnatal period in the US [14]. It was adapted to the European context in the framework of the “Research Innovation and Sustainable Pan-European Network in Peripartum Depression Disorder—Riseup-PPD” (Cost Action 18138), to which the Istituto Superiore di Sanità, ISS (Italian National Institute of Health) and the Società Marcé Italiana per la salute mentale perinatale (Italian Marcé Society) took part for Italy [15]. A national group of 12 perinatal health experts coordinated by the ISS developed the COPE-IS Italian adaptation, which addresses the following issues: perinatal health care experiences related to the COVID-19 pandemic (pregnant women: 20 items; women in the postnatal period: 32 items); COVID-19 exposure and symptoms (6 items); COVID-19 financial impact (1 item); COVID-19 social support impact and activity restrictions (8 items); COVID-19 coping strategies (24 items); COVID-19 emotional impact (12 items); physical and mental health history and substance use (4 items). A pilot test with a small sample of four pregnant women and four new mothers was performed to assure comprehensibility of the items.

The Brief Symptom Inventory-18 (BSI-18), omitting suicidality, was used to evaluate psychological distress in the last week, accordingly to previous pre-pandemic and pandemic studies involving pregnant women and new mothers [16,17]. The BSI-18 [18], which covers 18 of the 90 items in the SCL-90-R [19], is a self-report questionnaire including a summary scale of overall distress, the Global Severity Index (GSI), and three subscales: depression, anxiety and somatization. Each of the 18 items is rated on a 5-point Likert scale, with distress ratings ranging from 0 (not at all) to 4 (extremely). We used the validated Italian version [20].

A subset of 12 socio-demographics questions about the women’s year and country of birth, region of residency, educational level, number of children, number of people living at home, marital status, cohabitation with their partner, characteristics of the house, living environment changes since the beginning of the pandemic and financial situation was adapted from previous national studies on perinatal women coordinated by the ISS [21], and it completed the study instruments.

The whole questionnaire (COPE-IS Italian version, BSI-18 and demographic questions) took approximately 20 min to complete.

### 2.3. Sample Size Calculation

We estimated a minimum sample size of 400 pregnant women and 400 in the postnatal period based on an α-level of 0.05 and heterogeneity equal to 50%. We did not set any restriction on the number of subjects to be enrolled at the FCC level.

### 2.4. Statistical Methods

Pregnant women and those in the postnatal period were analyzed separately. A description of socio-demographic characteristics; anamnestic risk factors; women’s experience of support received from maternity services and belonging to a low (COVID-19 age-standardized mortality rates per 100,000 in 2020 ≤50), medium (> 50, < 100) or high (>100) [22] diffusion area for COVID-19 was reported for both groups. A description of further resources that the health services should provide, according to the users’ perspectives, was reported by the perinatal period and COVID-19 risk area. A comparison was made through a chi2 test or Fisher exact test.

The main outcome of interest was the psychological distress (GSI) obtained as the sum of the BSI-18 items, omitting suicidality, due to the online format of this study. The internal consistency of the BSI-18 was first assessed through the Cronbach’s alpha coefficient. The GSI was analyzed as a dichotomous variable; a cut-off of 25 was used, corresponding to a normalized T-score of 63, which is recommended to detect the prevalence of clinically relevant distress symptoms [20]. To further investigate the symptoms of discomfort in the period of pregnancy compared to the postnatal period, comparisons were made through the chi2 tests, with respect to the GSI and the three subscales that compose it: depression, anxiety and somatization. Logistic regression models were used to estimate the association between clinically relevant distress symptoms (GSI > 25) and potential risk factors. Adjusted Odds Ratios (aORs) and related 95% Confidence Intervals (CI) were estimated for the main sociodemographic (age, citizenship, education, financial condition), clinical (previous pregnancies, previous mood or anxiety disorders), perceived support (perceived social support, perceived support delivered by health professionals during pregnancy or the postnatal period, respectively) and COVID-19 exposure (previous SARS-CoV-2 infection, death of a family member/close friend from COVID-19, COVID-19 diffusion in the area of residence) characteristics as the independent variables, with GSI as the dependent variable. No substantial evidence of multicollinearity was found among the variables entered into the models. To verify the robustness of the coefficients estimated by the regression models, the analyses were repeated after excluding subjects who reported a previous mood or anxiety disorder (168 women in pregnancy; 136 in the postnatal period). The STATA statistical software, version 13 (StataCorp, College Station, TX; USA), was used for all analyses.

## 3. Results

From 1 October 2020 to 31 May 2021, 2633 women accessed the online survey, answering at least one question; 525 women filled in only preliminary questions and were excluded. Overall, 1168 pregnant women (73 in the first, 395 in the second and 694 in the third trimester and 6 missing) and 940 women in the six months after giving birth (mean age of the child: 2.7 months (SD = 1.98)) were included in the analyses.

### 3.1. Sociodemographic and Clinical Characteristics

The mean age was 33.1 (SD = 4.68) among pregnant women and 33.8 (SD = 4.57) among those enrolled in the postnatal period. Table 1 summarizes the main sociodemographic and clinical characteristics of the participants. Over 90% were married or living with their partner and had Italian citizenship. The majority were highly educated, employed, without financial difficulties, primiparous and without any previous obstetric complications (Table 1). A total of 21.9% (N = 206) of women in the postnatal period underwent a caesarean section (CS), and 85.1% (N = 800) were breastfeeding at the time of survey participation. A previous mood or anxiety disorder had been experienced by 14.4% and 14.5% of participants in the pregnant and postnatal group, respectively.

### 3.2. COVID-19 Exposure and Perceived Support during the COVID-19 Pandemic

Table 2 describes COVID-19 exposure and level of support from a social network and health professionals during the COVID-19 outbreak, as perceived by women in the perinatal period. The direct exposure to SARS-CoV-2 infection was 5.9% in the pregnant and 5.0% in the postnatal group, while 6.5% of pregnant women and 8.1% of those in the postnatal period suffered the loss of a family member or a close friend from COVID-19. Participants in the postnatal group (75%) were more likely to live in an area with a high COVID-19 diffusion (northern Italian region) than participants in the pregnant group (50%). About a third of the participants felt unsupported by their social network during the pandemic (32.1% of pregnant women, 38.1% of postnatal women), retrospectively, compared to 16.8% (N = 196) and 20.0% (N = 197) of pregnant and postnatal women in the pre-pandemic period. A total of 9.3% of pregnant women and 23.2% of women within six months from childbirth did not feel well-supported by health professionals during the pandemic.

### 3.3. Perinatal Experiences during the COVID-19 Pandemic

Among pregnant women, 61.7% (N = 721) noticed changes in the way antenatal care was delivered due to the outbreak of COVID-19; more specifically, 19.7% (N = 230) could not visit the maternity unit before giving birth, while 7.2% (N = 84) reported exclusion of their partner from prenatal and fetal ultrasound scan appointments. More than half of the pregnant women (60.4%; N = 706) expressed concern about the child’s health and 80.9% (N = 945) about the possible absence of the partner or of another person of support during delivery as a result of the COVID-19 restrictive measures.

Among women in the postnatal period, 21.2% (N = 199) faced delivery alone; more specifically, 16.4% (N = 116) of them had residency in high COVID-19 diffusion areas (N = 707) and 35.6% (N = 83) in the medium or in the low COVID-19 diffusion areas (N = 233). Overall, 8.6% (N = 81) experienced reduced provision of analgesia during childbirth, and 7.0% (N = 66) were separated from the child immediately after delivery. Lack of breastfeeding or other postnatal support after discharge from the hospital was described by 20.1% (N = 189) of the mothers, while 15.2% (N = 143) were unable to discuss issues related to their mood with a health professional. Lastly, 48.0% (N = 451) of postnatal women expressed concern about the child’s health and 62.2% (N = 585) about how to care for the baby because of the COVID-19 outbreak.

### 3.4. Resources Perceived as Helpful during the COVID-19 Outbreak

Table 3 shows the percentages of women who rated as important or very important the resources listed in the COPE-IS that helped them and their families during the pandemic, classifying the results according to perinatal period and COVID-19 diffusion in the area of residence of the participants. A rapid response to questions and concerns and wider availability of individual talks with health professionals caring for the perinatal woman and for the child were seen as important/very important by almost all participants, with proportions between 97.8% and 94.6%. The majority of women, both in pregnancy and in the postnatal group, also rated as important/very important having access to information on stress management (91.2% and 93.1%, respectively); a mental health professional (82.8% and 88.6%) and peer support, including online support groups (79.0% and 80.7%), interactions with other pregnant women/parents (91.6% and 93.7%) and experiences of women facing changes related to the perinatal period (83.7% and 84.6%).

Considering the COVID-19 diffusion throughout the Country, receiving information about stress management was considered important/very important, especially by pregnant women living in a low COVID-19 diffusion area (96.8%), followed by participants living in the medium (91.7%) and high (89.3%; *p* = 0.007) diffusion areas. The same pattern was observed in the postnatal group as for the greater availability of individual clinical talks with health professionals caring for postnatal women, while interactions with other women facing changes related to the perinatal period and with other parents were evaluated as less important in the medium COVID-19 diffusion Regions of residence compared to other areas (Table 3).

### 3.5. Psychological Distress

Pregnant women had a mean BSI-18 GSI score of 11.75 (SD = 9.89) compared to a mean of 10.69 (SD = 9.71) among women in the postnatal period (*p* = 0.02). A higher proportion of participants with a GSI score > 25, which identifies clinically relevant distress symptoms, was found in the pregnant (12.1%) compared to the postnatal group (9.3%; *p* = 0.038) (Table 4). As for the BSI-18 subscales, pregnant women showed a higher prevalence of symptoms of somatization (6.9%) than those in the postnatal period (2.6%, *p* < 0.001). No other statistically significant differences were found.

In Table 5, the adjusted ORs and related 95% CIs are reported to evaluate the independent association of sociodemographic, clinical and COVID-19 exposure characteristics with clinically relevant psychological distress symptoms among women in pregnancy and during the postnatal period. The number of records having valid values for all the variables included in the model were 1154 in the pregnant and 915 in the postnatal group. Pregnant women with clinically relevant psychological distress symptoms (GSI score > 25) were more likely to experience financial difficulties (aOR: 2.87; 95% CI: 1.92–4.29), to have had a previous mood or anxiety disorder (aOR: 3.55; 95% CI: 2.31–5.45), to report a lack of perceived social support (aOR: 1.75; 95% CI: 1.18–2.61) and of support provided by health professionals during pregnancy (aOR: 2.31; 95% CI: 1.35–3.94) and to have faced the death of a family member or close friend from COVID-19 (aOR: 3.36; 95% CI: 1.85–6.09) than pregnant women with a GSI score <25. Among women in the postnatal period, those with relevant distress symptoms compared to those with a GSI score <25 were less frequently multiparous (aOR: 0.50; 95% CI: 0.25–0.93) and were more likely to have financial difficulties (aOR: 1.87; 95% CI: 1.12–3.12), to have experienced a previous mood or anxiety disorder (aOR: 4.12; 95% CI: 2.45–6.91) and to suffer a lack of perceived social support (aOR: 2.78; 95% CI: 1.69–4.57) and of support provided by health professionals during the postnatal period (aOR: 2.59; 95% CI: 1.57–4.27). The coefficients estimated by the logistic regression models after excluding subjects with a previous mood or anxiety disorder among pregnant women and those in the postnatal period showed minor variations. All associations of the independent variables with psychological distress were confirmed, with the exception of the variable “lack of perceived social support” among pregnant women, which did not reach statistical significance (OR = 1.46; *p* = 0.116).

## 4. Discussion

This has been the first study describing the characteristics associated with psychological distress among women in the perinatal period accessing the Italian community-based primary care services dedicated to women’s health (i.e., the FCCs). Financial difficulties, a previous mood or anxiety disorder and a lack of perceived social support and support provided by health professionals were associated with relevant psychological distress symptoms, both among pregnant women and those in the postnatal period. No association was found between direct exposure to SARS-CoV-2 infection or residence in a high COVID-19 diffusion area and mental health status.

The overall prevalence of relevant psychological distress of about 12% among pregnant women and 9% among those in the postnatal period is consistent with rates observed in a large multinational European web-based study (anxiety symptoms among 11% of pregnant and 10% of breastfeeding women, as measured by the Generalized Anxiety Disorder 7-items scale; depressive symptoms among 15% and 13%, respectively, as measured by the Edinburgh Depression Scale) conducted beyond the peak of the first wave of the epidemic on a sample presenting similar socio-demographic features [23]. Otherwise, online cross-sectional surveys based on small samples of perinatal women in contact with single university centers located in northern [24,25], central [26] and southern Italy [27] registered clinically significant anxiety among 64.0% to 77.0% [24,26,27] of pregnant women and 57.7% of those in the postnatal period [24] and significant symptoms of depression among 26.3% [24] to 44.2% [25] of women within 6 months after childbirth assessed during the pandemic’s first wave. The lower prevalence of psychological distress in our sample compared to previous Italian estimates could partly be explained by differences in sample size and characteristics, instruments adopted for mental health assessment and by the timing and specific settings of recruitment. Samples recruited during the Italian first wave included 100 to 178 participants [25,26,27], except for one study that was based on 388 pregnant and 186 postpartum women [24], who showed, however, a much higher prevalence of previous psychological disorders (>50%) and complications during pregnancy (40%) compared to the 14% and 21% we registered, respectively. While the above-mentioned studies used the State-Trait Anxiety Inventory to assess anxiety symptoms [24,27] and the Edinburgh Postnatal Depression Scale–EPDS to assess depression symptoms [24,25], our results were based on the BSI-18, which could have been less sensitive to symptoms of depression and anxiety among women in the perinatal period. It has also been suggested that prenatal anxiety increased with the severity of the restriction measures [28], which were higher in the first Italian epidemic wave compared to the second, when our study was conducted. Moreover, activities offered to perinatal women by the FCCs, widely made available remotely at the time of the study [13,29], possibly contributed to mitigate the psychosocial effect of the pandemic in our sample compared to women with an undefined care pathway, recruited via social-media or in contact with tertiary hospitals facing the huge burden of COVID-19 disease [30] and caring mostly for patients with complicated pregnancy that could be more prone to higher levels of distress related to the obstetric risks [31]. However, pre-pandemic studies conducted among FCC users found depressive symptoms among 7.4% and depressive and anxiety symptoms among 13% of women assessed on EPDS [32] and on EPDS and GHQ-12 [33] 6 to 12 weeks after childbirth, respectively, while women with university qualifications were found to be less than half as likely to suffer from a high level of depressive symptomatology after childbirth [33]. Since highly educated women represented 58% of our sample, the prevalence of relevant depression (13.7%) and anxiety symptoms (17.3%) in our study was probably higher compared to historical FCC controls.

The higher psychological distress among pregnant women compared to those in the postnatal period we observed appeared to be mostly related to the high somatization scores in the pregnant group (Table 5). It should be mentioned that the BSI-18 somatization subscale does not evaluate the presence of somatic symptoms (which could be due to pregnancy) but how they are perceived or experienced, which was identified as a source of significant psychological distress, particularly in the group of pregnant women in our study, and this was associated with depressive and anxiety disorders during pregnancy in previous research [34].

The association between low income and a history of depression or anxiety with a higher risk of psychological distress among women in the perinatal period is well recognized [35] and is consistent with findings of other studies conducted in the last months [22,29]. The high percentage of pregnant women (32.1%) and of those in the postnatal period (38.1%) reporting a lack of social support should be a cause of concern, due to its crucial role in the physical, mental and emotional well-being of mothers [36], also reaffirmed in the COVID-19 emergency [23,24].

Support provided by health professionals to women in the perinatal period and health service ability to listen to women’s needs is of pivotal importance in the present unexpected and constantly evolving scenario. The protective role of a higher perceived level of healthcare support against psychological distress during the pandemic is consistent with other findings [25]. Pregnant women in our study were worried about change in the maternity care, and most of them expressed concern about the absence of their partners during delivery as a result of the COVID-19 restrictive measures, which has been found to predict a women’s mental health [24]. Despite the low rate of SARS-CoV-2 positivity (5%) in our sample, on average, 21.3% of the women in the postnatal period faced delivery alone. The exclusion of a partner from childbirth was less common in high COVID-19 diffusion areas compared to medium or low areas, suggesting that a better preparedness had been developed by the maternity services located in the Italian geographical areas with a high COVID-19 mortality rate, those most affected by the first wave, compared to the others. Participants identified three main areas that could help them and their family during the pandemic: improved access to medical providers, mental health care and self-help resources. More specifically, 1:5 woman experienced a lack of postnatal support after discharge from the hospital, including the impossibility to disclose emotions and their mood to a health professional. According to FCC users, greater support is required in the first months after childbirth compared to resources offered during pregnancy.

Different from the potential risk for mental health among participants with a high direct and indirect COVID-19 exposure that we had envisaged, SARS-CoV-2 infection and residence in a high COVID-19 diffusion area were not associated with psychological distress, either among pregnant women or among those after childbirth, in agreement with previous international [23] and national findings [37]. It should be mentioned that the BSI-18 only collects information over the last week and that the timing of the infection was not available; therefore, we were not able to exclude high distress in the immediacy of the infection. The disproportionate adoption of measures restricting health care access compared to COVID-19 diffusion in the same area could have contributed to the lack of an association observed in our sample.

The loss of a beloved one from COVID-19 was associated with psychological distress among pregnant women. While the impact of COVID-19-related grief on mental health symptoms and maternal–infant bonding in the postnatal period has been recognized [38], our results suggest the need for taking into account losses experienced during pregnancy.

Our findings should be interpreted in light of two main limitations. First, while recruitment took place in the FCCs, online participation may have introduced risks of recall and selection bias, inherent to web surveys [39]. Accordingly, highly educated and Italian women in our sample were overrepresented compared with figures reported by the National Birth Register [40], advising for caution when generalizing our findings to all women in the perinatal period. Participants’ sociodemographic characteristics were rather similar to those of women attending FCC perinatal classes [41,42], which have been recognized as a group with a better health status compared to the general population of pregnant and postpartum women [32]. Second, we acknowledge the limitation in the use of a non-validated measure, such the COPE-IS, to assess the areas of impact of COVID-19 and of the BSI-18 to evaluate psychological distress, which due to the recent Italian validation, could not rely on national pre-pandemic controls. On the other hand, since both the COPE-IS and the BSI-18 have been adopted by contemporary studies conducted among perinatal women in 11 European countries [15], our study will provide valuable findings for international comparisons during the COVID-19 outbreak.

## 5. Conclusions

As psychological distress produces adverse maternal and infant outcomes, our study highlighted the urgency to provide enhanced care to the most vulnerable women who face pregnancy and the first months after childbirth in the present context, even independently of direct SARS-CoV-2 exposure. While clinically relevant psychological distress was observed in about 1 in 10 women, changes in maternity care and reduced social support related to the COVID-19 outbreak were causes of concern among the large majority of the participants, who asked for greater listening by health professionals, postnatal and mental health support and self-help resources. It is time to re-establish the centrality of women and the parity of esteem between physical and mental health and to partner involvement among the guiding principles of pregnancy, childbirth and postnatal care.

## Figures and Tables

**Table 1 ijerph-19-01983-t001:** Sociodemographic and clinical characteristics of the study population, according to the perinatal period.

	Women in Pregnancy		Women in the Postnatal Period	
	N = 1168	%	N = 940	%
Age				
≤30	310	26.5	207	22.0
31–37	648	55.5	517	55.0
≥38	197	16.9	192	20.4
Missing	13	1.1	24	2.6
Marital status				
Married or living with partner	1137	97.3	910	96.8
Single/Separated/Divorced/Widowed	20	1.7	6	0.6
Missing	11	0.9	24	2.6
Citizenship				
Italian	1080	92.5	879	93.5
Not Italian	75	6.4	36	3.8
Missing	13	1.1	25	2.7
Education				
Low	497	42.6	346	36.8
High (Bachelor’s degree or higher)	658	56.3	570	60.6
Missing	13	1.1	24	2.6
Employment status				
Employed	918	78.6	761	81.0
Unemployed	185	15.8	113	12.0
Housewife, student	65	5.6	66	7.0
Financial difficulties				
No	812	69.5	651	69.3
Yes	342	29.3	265	28.2
Missing	14	1.2	24	2.6
Parity				
Primiparous	896	76.7	641	68.2
Multiparous	259	22.2	275	29.3
Missing	13	1.1	24	2.6
Obstetrics complications ^1^				
No	904	77.4	630	67.0
Yes	246	21.1	303	32.2
Missing	18	1.5	7	0.7
Previous mood and/or anxiety disorder				
No	1000	85.6	804	85.5
Yes	168	14.4	136	14.5

^1^ Obstetric complications: complication during pregnancy for pregnant women or complications during pregnancy and childbirth for women in the postnatal period.

**Table 2 ijerph-19-01983-t002:** COVID-19 exposure and perceived support by a social network and health professionals during the COVID-19 outbreak, according to women in the perinatal period.

	Women in Pregnancy		Women in the Postnatal Period	
	N = 1168	%	N = 940	%
SARS-CoV-2 infection				
no	1099	94.1	893	95.0
yes	69	5.9	47	5.0
Death of a family member/close friend from COVID-19				
no	1092	93.5	864	91.9
yes	76	6.5	76	8.1
COVID-19 diffusion in the area ^1^ of residence				
low diffusion	157	13.4	100	10.6
medium diffusion	421	36.0	133	14.2
high diffusion	590	50.5	707	75.2
Perceived social support				
Supported to very well supported	788	67.5	582	61.9
Unsupported	380	32.5	358	38.1
Perceived support provided by health professionals				
Somewhat well/very well supported	1059	90.7	722	76.8
Not very well supported	109	9.3	218	23.2

^1^ COVID-19 diffusion in the area of residence: low diffusion = COVID-19 age-standardized mortality rates per 100,000 in 2020 ≤ 50; medium diffusion = COVID-19 mortality rates per 100,000 >50 ≤100; high diffusion = COVID-19 mortality rates per 100,000 > 100.

**Table 3 ijerph-19-01983-t003:** Proportion of women who considered the listed resources to be important in the pandemic context, according to perinatal period and COVID-19 diffusion area.

Resources	All	Residents in a Low COVID-19 Diffusion Area	Residents in a Medium COVID-19 Diffusion Area	Residents in a High COVID-19 Diffusion Area	
	**Pregnant women**				
	N = 1168	N = 157	N = 421	N = 590	
	%	%	%	%	*p*
Rapid response to questions and concerns	97.8	98.1	98.1	97.5	0.79
Greater availability of individual talks with pregnancy healthcare professionals	96.8	98.7	96.4	96.6	0.35
Interactions with other pregnant women	91.6	93.6	92.6	90.3	0.27
Information on stress management	91.2	96.8	91.7	89.3	0.01
Access to experiences of women facing changes related to the perinatal period	83.7	82.8	85.5	82.7	0.47
Access to a mental health professional	82.8	87.9	82.7	81.5	0.17
Online support groups	79.0	84.7	79.1	77.5	0.14
	**Women in the postnatal period**				
	N = 940	N = 100	N = 133	N = 707	
	%	%	%	%	*p*
Rapid response to questions and concerns	98.6	97.0	97.7	99.0	0.10
Greater availability of individual talks with health professionals caring for postnatal women	94.6	100.00	96.2	93.5	<0.01
Greater availability of individual talks with the pediatrician	97.3	98.0	98.5	97.0	0.78
Interactions with other parents	93.7	96.0	88.0	94.5	0.02
Information on the COVID-19 impact on infant/child health	98.1	100.00	97.0	98.0	0.28
Information on stress management	93.1	97.0	93.2	92.5	0.26
Access to experiences of women facing changes related to the perinatal period	84.6	92.0	80.0	84.6	0.02
Access to a mental health professional	88.6	92.0	85.7	88.7	0.32
Online support groups	80.7	82.0	73.7	81.9	0.08

**Table 4 ijerph-19-01983-t004:** BSI-18 Global Short Index (GSI) and subscales scores according to the perinatal period.

	Women in Pregnancy	Women in the Postnatal Period	Total	X^2^ Test *p*
N	1168	940	2108	
BSI-18 GSI				
% GSI > 25	12.1	9.3	10.8	0.04
Somatization				
% Somatisation Score > 8	6.9	2.6	5.0	<0.01
Depression				
% Depression Score > 8	12.8	14.7	13.7	0.22
Anxiety				
% Anxiety Score > 8	18.1	16.4	17.3	0.31

**Table 5 ijerph-19-01983-t005:** Logistic regression of factors associated with clinically relevant distress symptoms among women in pregnancy or in the postnatal period.

**Pregnant Women**			
GSI ≥ 25	**Adjusted OR**	**95% CI**	
Age 31–37 years (vs <30)	1.48	0.90	2.43
Age ≥ 38 years (vs. <30)	1.27	0.67	2.40
Multiparous (vs primiparous)	1.05	0.66	1.65
Not Italian citizenship (vs. Italian)	0.62	0.24	1.62
Highly educated (vs. low)	0.91	0.60	1.36
Financial difficulties (yes vs. no)	2.87	1.92	4.29
Previous mood or anxiety disorder (yes vs. no)	3.55	2.31	5.45
Lack of perceived social support (unsupported vs. well/very well supported)	1.75	1.18	2.61
Perceived lack of support by health professionals during pregnancy (not very well supported vs. somewhat well/very well supported)	2.31	1.35	3.94
SARS-CoV-2 infection (yes vs. no)	0.62	0.24	1.61
Death of a family member/close friend from COVID-19 (yes vs. no)	3.36	1.85	6.09
Residence in a medium COVID-19 diffusion area (vs. low)	1.31	0.69	2.49
Residence in a high COVID-19 diffusion area (vs. low)	1.20	0.64	2.25
**Women in the postnatal period**			
GSI > 25	**Adjusted OR**	**95% CI**	
Age 31–37 years (vs. <30)	1.04	0.57	1.91
Age ≥ 38 years (vs. <30)	0.99	0.46	2.12
Multiparous (vs. primiparous)	0.50	0.27	0.93
Not Italian citizenship (vs Italian)	1.84	0.59	5.76
Highly educated (vs. low)	0.89	0.53	1.50
Financial difficulties (yes vs. no)	1.87	1.12	3.13
Previous mood or anxiety disorder (yes vs. no)	4.12	2.45	6.91
Lack of perceived social support (unsupported vs. well/very well supported)	2.78	1.69	4.57
Perceived lack of support by health professionals in the postnatal period (not very well supported vs. somewhat well/very well supported)	2.59	1.57	4.27
SARS-CoV-2 infection (yes vs. no)	1.14	0.41	3.21
Death of a family member/close friend from COVID-19 (yes vs. no)	1.47	0.69	3.14
Residence in a medium COVID-19 diffusion area (vs. low)	0.84	0.29	2.42
Residence in a high COVID-19 diffusion area (vs. low)	1.46	0.67	3.18

## Data Availability

Not applicable.

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
