# Peer review of "The Psychological Impact of COVID-19 among Women Accessing Family Care Centers during Pregnancy and the Postnatal Period in Italy"

_ijerph, 2022, doi:10.3390/ijerph19041983_

Round 1
Reviewer 1 Report
I have read this paper with great interest.
This paper has its focus on psychosocial impact and well-being of pregnant and postpartum women during the covid pandemic. The WHO-MNCAH research prioritization group recently published on the research priorities on COVID-19for maternal, newborn, child and adolescent health (J Glob Health 2021, PMID 34912548), and this includes the psychosocial aspects. Likely valuable to put your paper into this broader perspective.
The methods and the questionnaire are valid.
There is another paper in this journal reporting on a questionnaire during pregnancy and postpartum in the same time interval of the Italian study, so that it is interesting to compare these results with (Ceulemans et al, Int J Environ Res Public Health 2020, PMID 32957434)
Based on the results of the questionnaire (table 1), can the authors somewhat more reflect on the representability of their sample to the full population (eg age, marital status, educational, primi/mulitparous) of pregnant and postpartum women in their country ?
Author Response
Reviewer 1.
The WHO-MNCAH research prioritization group recently published on the research priorities on COVID-19 for maternal, newborn, child and adolescent health (J Glob Health 2021, PMID 34912548), and this includes the psychosocial aspects. Likely valuable to put your paper into this broader perspective.
We thank the reviewer for his/her valuable suggestion that we included in the Introduction as follows:
“The need for evidence focusing on the indirect effects of the COVID-19 pandemic on health of women in the perinatal period, including psychosocial aspects, and on measures to improve access and provision of care, including psychosocial assistance, has been authoritatively affirmed (COVID-19 Research Prioritization Group on MNCAH. Global research priorities on COVID-19 for maternal, newborn, child and adolescent health. J Glob Health. 2021 Nov 20;11:04071. doi: 10.7189/jogh.11.04071. PMID: 34912548; PMCID: PMC8645217)”
There is another paper in this journal reporting on a questionnaire during pregnancy and postpartum in the same time interval of the Italian study, so that it is interesting to compare these results with (Ceulemans et al, Int J Environ Res Public Health 2020, PMID 32957434)
The comparison with the results of the multinational results by Ceulemans M et al. (Mental health status of pregnant and breastfeeding women during the COVID-19 pandemic- A multinational cross-sectional study. Acta Obstet Gynecol Scand 2021;100(7):1219-1229) was included in the Discussion of the first version of our manuscript. We added a reference to the Belgian arm of the study published on the International Journal of Environmental Research and Public Health: reference n. 24 - Ceulemans M et al. SARS-CoV-2 Infections and Impact of the COVID-19 Pandemic in Pregnancy and Breastfeeding: Results from an Observational Study in Primary Care in Belgium. Int J Environ Res Public Health 2020; 17(18):6766.
Based on the results of the questionnaire (table 1), can the authors somewhat more reflect on the representability of their sample to the full population (eg age, marital status, educational, primi/mulitparous) of pregnant and postpartum women in their country ?
We thank the reviewer for this comment highlighting that the issue required clarification. We included the available national figures in the discussion as follows: “Participants were slightly older, higher educated, more professionally active and more often of Italian citizenship and living with partner, compared to national population data (age at childbirth= 33.0 years among Italian 30.8 among not Italian women; Bachelor’s degree or higher= 33%; employed= 56%; Italian citizenship=79%; married=61%) [39]. Therefore, caution is required when generalizing our findings to all women in the perinatal period in Italy.”
Reviewer 2 Report
Dear Editor:
Thank you for inviting me to review the article--The psychological impact of COVID-19 among women accessing Family Care Centres during pregnancy and the postnatal period in Italy.
In this article, the authors investigated that women in Italy during pregnancy and in the first year after childbirth have to face various stress factors during the covid-19 epidemic. It leads to maternal psychological distress. Through online surveys of maternity, assessment, comparison, and recording of maternal mental health status, no association was found between direct exposure to SARS-CoV-2 infection or residence in high COVID-19 diffusion area and mental health status . It has significant differences in which As psychological distress produce adverse maternal and infant outcomes, our study highlights the urgency to provide enhanced care to the most vulnerable women who face pregnancy and the first months after childbirth in the present context, even independently of direct SARS-CoV-2 exposure. It’s time to re-establish centrality of women, parity of esteem between physical and mental health and partner involvement among the guiding principles of pregnancy, childbirth and the postnatal care . Overall, the article is well organized and its presentation is good. Here are the comments on this article:
Major Comments#1: Inclusion criteria: women with mental illness were not excluded.
Major Comments#2: Whether there is collinearity between their variables in logistic regression analysis.
Major Comments#3:The influence of factors on stress was not verified, and please further exclude the patients with prenatal depression, then perform sensitivity analysis.
From the above, it is recommended to accept after revision.
Author Response
Major Comments#1: Inclusion criteria: women with mental illness were not excluded.
Major Comments#3: The influence of factors on stress was not verified, and please further exclude the patients with prenatal depression, then perform sensitivity analysis.
Our aim was to evaluate the psychological impact - in terms of psychological distress - of the COVID-19 pandemic on all the women accessing Family Care Centres during pregnancy or in the postnatal period, in a “real world” perspective. However, we are in complete agreement with the reviewer on the need for conducting an adjunctive analysis excluding women with previous mood or anxiety disorder to limit the possible confounding effect of disease relapses. We could not exclude patients with prenatal depression as requested, as we do not have this information. We repeated the regression analyses after excluding subjects with previous mood or anxiety disorder.
We added the following phrase in the Methods section: “To verify the robustness of the coefficients estimated by the regression models, the analyses were repeated after excluding subjects who reported previous mood or anxiety disorder (168 women in pregnancy and 136 women in the postnatal period).”
In the Results: “The coefficients estimated by the logistic regression models after excluding subjects with previous mood or anxiety disorder among pregnant women and those in the postnatal period showed minor variations. All associations of the independent variables with psychological distress were confirmed with the exception of the variable “lack of perceived social support” among pregnant women, which did not reach statistical significance (OR = 1.46; p = 0.116).”
Major Comments#2: Whether there is collinearity between their variables in logistic regression analysis.
As suggested, independent variables in the models were explored to rule out the presence of multicollinearity. Correlation among independent variables, correlation of the estimated coefficients of the logistic models and the Variance Inflaction Factors (VIF) showed no signs of collinearity: all Variance Inflaction Factors (VIF) were <4. We added the following sentence in the methods section: “No substantial evidence of multicollinearity was found among variables entered the models”.